# Measuring protein stability in the GroEL chaperonin cage reveals massive destabilization

**Ilia Korobko[1], Hisham Mazal[2], Gilad Haran[2], Amnon Horovitz[1]\***

[1]Departments of Structural Biology, Weizmann Institute of Science, Rehovot, Israel; [2]Chemical and Biological Physics, Weizmann Institute of Science, Rehovot, Israel

**Abstract** The thermodynamics of protein folding in bulk solution have been thoroughly investigated for decades. By contrast, measurements of protein substrate stability inside the GroEL/ES chaperonin cage have not been reported. Such measurements require stable encapsulation, that is no escape of the substrate into bulk solution during experiments, and a way to perturb protein stability without affecting the chaperonin system itself. Here, by establishing such conditions, we show that protein stability in the chaperonin cage is reduced dramatically by more than 5 kcal mol$^{-1}$ compared to that in bulk solution. Given that steric confinement alone is stabilizing, our results indicate that hydrophobic and/or electrostatic effects in the cavity are strongly destabilizing. Our findings are consistent with the iterative annealing mechanism of action proposed for the chaperonin GroEL.

## Introduction

The *Escherichia coli* GroE chaperonin system, which comprises GroEL and its co-factor GroES, assists protein folding in vivo and in vitro in an ATP-dependent manner (*Thirumalai and Lorimer, 2001*; *Saibil et al., 2013*; *Hayer-Hartl et al., 2016*; *Gruber and Horovitz, 2016*). Binding of GroES to GroEL forms a cage in which encapsulated substrate proteins can fold in isolation from bulk solution. The GroE system has been studied intensively for more than three decades, but it is still unclear and controversial whether its cavity is a 'passive cage' in which protein substrate aggregation is prevented but the folding pathway is unchanged or a chamber in which the folding process is altered in some manner. It is also unclear whether encapsulation in the GroE cavity is thermodynamically stabilizing, for example because of confinement, or destabilizing owing, for example, to a diminished hydrophobic effect.

The effects of encapsulation on folding kinetics have been examined in different studies but it has been difficult to generalize its impact, in part, because it may depend on the properties of the protein substrate studied such as its size and charge. According to some studies, encapsulation prevents aggregation but does not alter the protein substrate's folding pathway and kinetics (*Horst et al., 2007*; *Tyagi et al., 2011*). In the case of rhodanese, for example, it was reported that encapsulation retards the folding of its C-terminal domain but has no effect on its N-terminal domain (*Hofmann et al., 2010*). By contrast, other studies have suggested more 'active' models according to which folding is accelerated in the GroEL cavity compared to bulk solution. Recent work showed, for example, that GroEL repairs a folding defect of a mutant of maltose-binding protein by accelerating the formation of an on-pathway intermediate (*Ye et al., 2018*). Accelerated folding upon encapsulation has been attributed to steric confinement and the negative charges of the cavity walls (*Tang et al., 2006*). It has also been attributed to substrate interactions with the cavity-protruding C-terminal tails of GroEL, which comprise Gly-Gly-Met repeats (*Tang et al., 2006*; *Weaver and Rye, 2014*; *Weaver et al., 2017*). In addition, GroEL-mediated folding can be affected by the cavity-

***For correspondence:**
Amnon.Horovitz@weizmann.ac.il

**Competing interests:** The authors declare that no competing interests exist.

**eLife digest** All cells contain molecules known as proteins that perform many essential roles. Proteins are made of chains of building blocks called amino acids that fold to form the proteins' three-dimensional structures. Many proteins fold spontaneously into their well-defined and correct structures. However, some proteins fold incorrectly, which prevents them from working properly, and can lead to formation of aggregates that may harm the cell.

To prevent such damage, cells have evolved proteins known as molecular chaperones that assist in the folding of other proteins. For example, a molecular chaperone called GroEL is found in a bacterium known as *Escherichia coli*. This molecular chaperone contains a cavity which prevents target proteins from forming clumps by keeping them away from other proteins. However, it remained unclear precisely how GroEL works and whether enclosing target proteins in its cavity has other effects.

*Moritella profunda* is a bacterium that thrives in cold environments and, as a result, many of its proteins are unstable at room temperature and tend to unfold or fold incorrectly. To study how GroEL works, Korobko et al. used a protein from *M. profunda* called dihydrofolate reductase as a target protein for the chaperone. A clever trick was then used to determine the folding state of dihydrofolate reductase when inside the chaperone cavity. The experiments revealed that the environment within the cavity of GroEL strongly favors dihydrofolate reductase adopting its unfolded state instead of its folded state. This suggests that GroEL helps dihydrofolate reductase and other incorrectly folded target proteins to unfold, thus providing the proteins another opportunity to fold again correctly.

Parkinson's disease, Alzheimer's disease and many other diseases are caused by proteins folding incorrectly and forming aggregates. A better understanding of how proteins fold may, therefore, assist in developing new therapies for such diseases. These findings may also help biotechnology researchers develop methods for producing difficult-to-fold proteins on a large scale.

confined water, which can enhance the hydrophobic effect (and thus accelerate folding) by accumulating near the cavity walls (*England et al., 2008*) or diminish it (and thus slow folding) by being more ordered. Finally, an even more 'active' mechanism has been proposed according to which encapsulated misfolded protein substrates undergo ATP-promoted forced-unfolding (*Weaver and Rye, 2014*), in accordance with the iterative annealing model (*Todd et al., 1996*), thereby giving them further opportunity to fold correctly. The iterative annealing model predicts higher folding yields but it has been suggested that annealing can also lead to accelerated folding (*Tang et al., 2006*; *Gupta et al., 2014*; *Weaver and Rye, 2014*; *Weaver et al., 2017*).

In contrast with the many and often conflicting studies on the kinetic effects of encapsulation, there have been virtually no reports on the thermodynamic effects of encapsulation. One complication in understanding the thermodynamic effects of encapsulation has been that protein substrates spend variable amounts of time folding in the cavity and in bulk solution because of GroEL-GroES cycling and the leakiness of its cavity (*Motojima and Yoshida, 2010*). As shown below, such leakiness is even greater in the complex of single-ring GroEL with GroES, which was used in some studies (*Hofmann et al., 2010*) and has been suggested to form during GroEL's normal reaction cycle (*Yan et al., 2018*). Another complication has been that measuring protein stability usually involves perturbations such as temperature or solution changes, which could also affect the stability of the chaperonin itself. The goal of the work described here was, therefore, to establish a system that would allow measuring the stability of an encapsulated protein in a non-leaky and unperturbed GroE complex. An appropriate system was found to be a chimera of dihydrofolate reductase from *Moritella profunda* ($DHFR_{Mp}$) fused to eGFP, which is encapsulated in the football-shaped and non-cycling $BeF_x$-stabilized GroEL-GroES$_2$ complex (*Figure 1*). Consequently, we were able to isolate the thermodynamic effect of encapsulation from other effects associated with GroEL-GroES cycling. Our results show that protein stability in the GroEL cavity is reduced dramatically in comparison with bulk solution.

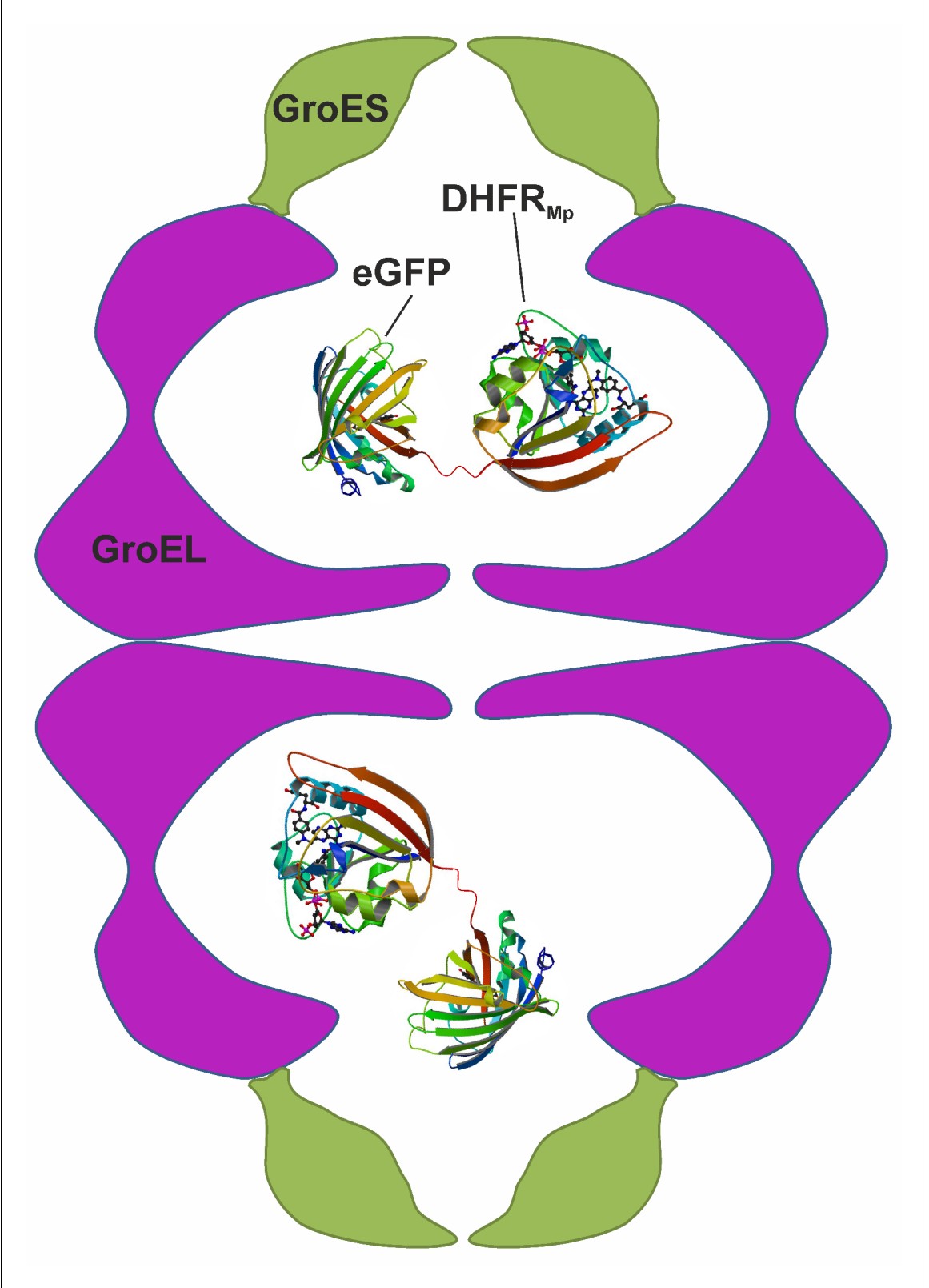

**Figure 1.** Scheme showing the experimental system used to measure protein stability in the chaperonin cage. A chimera was formed by fusing dihydrofolate reductase from *Moritella profunda* (DHFR_Mp, PDB ID: 3IA4) to the C-terminus of eGFP (PDB ID: 2Y0G) via a linker. This chimera was then encapsulated in the cavity of the BeF_x-stabilized football–shaped GroEL-GroES_2 complex, which is depicted schematically with GroEL in purple and
*Figure 1 continued on next page*

*Figure 1 continued*

GroES in green. The cavity volume is about 175,000 Å$^3$ whereas that of the chimera (which is not drawn to scale) is about 58,000 Å$^3$ assuming 1.23 Å$^3$ per Da.

The online version of this article includes the following figure supplement(s) for figure 1:

**Figure supplement 1.** Isolation and stability of eGFP-containing GroEL-GroES$_2$ football-shaped complexes.
**Figure supplement 2.** Isolation and stability of eGFP-containing single-ring GroEL in complex with GroES.

## Results

### Design and construction of a model protein substrate

We chose to use dihydrofolate reductase from *Moritella profunda*, a psychrophilic bacterium with an optimal growth temperature at 2°C, as a model substrate since it is a relatively unstable protein at room temperature *Xu et al., 2003* whose folding can be monitored easily by measuring the regain in enzyme activity. DHFR$_{Mp}$ was fused to the C-terminus of enhanced green fluorescent protein (eGFP) in order to further destabilize it as observed before for other proteins (*Sokolovski et al., 2015*; *Dave et al., 2016*). The fusion to eGFP also facilitated easy and sensitive determination of the location of the substrate, that is whether it is in the cavity or has leaked outside of it. Under our conditions, the apparent melting temperature, T$_m$, in bulk solution of DHFR$_{Mp}$ alone was found to be about 41.2 (±1.8) °C whereas DHFR$_{Mp}$ fused to eGFP was found to be strongly destabilized with a T$_m$ of 22.8 (±1.1) °C (*Figure 2*). eGFP in the chimera was also found to be destabilized in bulk solution relative to eGFP alone, but to a lesser extent, with T$_m$ values of 76.3 (±0.3) and 81.3 (±1.3) °C, respectively.

### Identifying conditions for minimizing escape of encapsulated substrates

GroEL and GroES can form either GroEL-GroES bullet-shaped or GroEL-GroES$_2$ football-shaped complexes. It has been reported that the GroEL-GroES$_2$ complex is stabilized in the presence of BeF$_x$ (*Taguchi et al., 2004*). We, therefore, tested whether eGFP remains encapsulated in the BeF$_x$-stabilized 'football' complex over a sufficiently long period of time. BeF$_x$-stabilized and eGFP-containing 'football' complexes were prepared, purified by gel-filtration and then allowed to stand overnight at room temperature. The samples were then analyzed by gel-filtration in order to determine the extent, if any, of substrate escape. The results show that all the eGFP co-eluted with GroEL and GroES, thereby indicating that the complex remained intact and no escape occurred (*Figure 1—figure supplement 1*). Two other proteins, the p53 core domain and a chimera of the engrailed home-odomain transcription factor with eGFP with respective masses of 22.4 and 42.9 kDa, were also tested in this manner and found to not escape (data not shown). Previously, it was believed that substrates encapsulated in the cage formed by single-ring GroEL in complex with GroES cannot escape since dissociation of GroES from one ring of GroEL is triggered by ATP binding to the opposite ring (*Rye et al., 1997*). Experiments carried out with eGFP encapsulated in the cavity of single-ring GroEL in complex with GroES showed, however, that massive substrate leakage took place (*Figure 1—figure supplement 2*).

### Folding of the DHFR$_{Mp}$ part of the chimera in the GroEL cavity

BeF$_x$-stabilized 'football' complexes containing the chimera were purified and the DHFR activity of the encapsulated chimera was monitored at 23°C (i.e. near the apparent T$_m$ of the fused DHFR$_{Mp}$ in bulk solution) by following the change in absorbance at 340 nm, upon addition of the substrates NADPH and dihydrofolate (DHF). The data show a lag phase followed by a linear phase (*Figure 3*) before activity starts to diminish owing to substrate depletion. Strikingly, the lag phase is absent when the chimera is free in bulk solution (*Figure 3*). One possible reason for the presence of the lag phase, in the case of the encapsulated chimera, is that the added substrates (DHF and NADPH) need to diffuse into the cavity. In such a case, the rate constant associated with the lag phase should increase with increasing substrate concentrations. Alternatively, the lag phase may reflect substrate (DHF and/or NADPH)-promoted folding of the DHFR$_{Mp}$ part of the chimera, which is destabilized in the cavity relative to bulk solution. Substrate (DHF and/or NADPH)-promoted folding of the DHFR$_{Mp}$ part of the chimera can take place via a mechanism of conformational selection, that is the

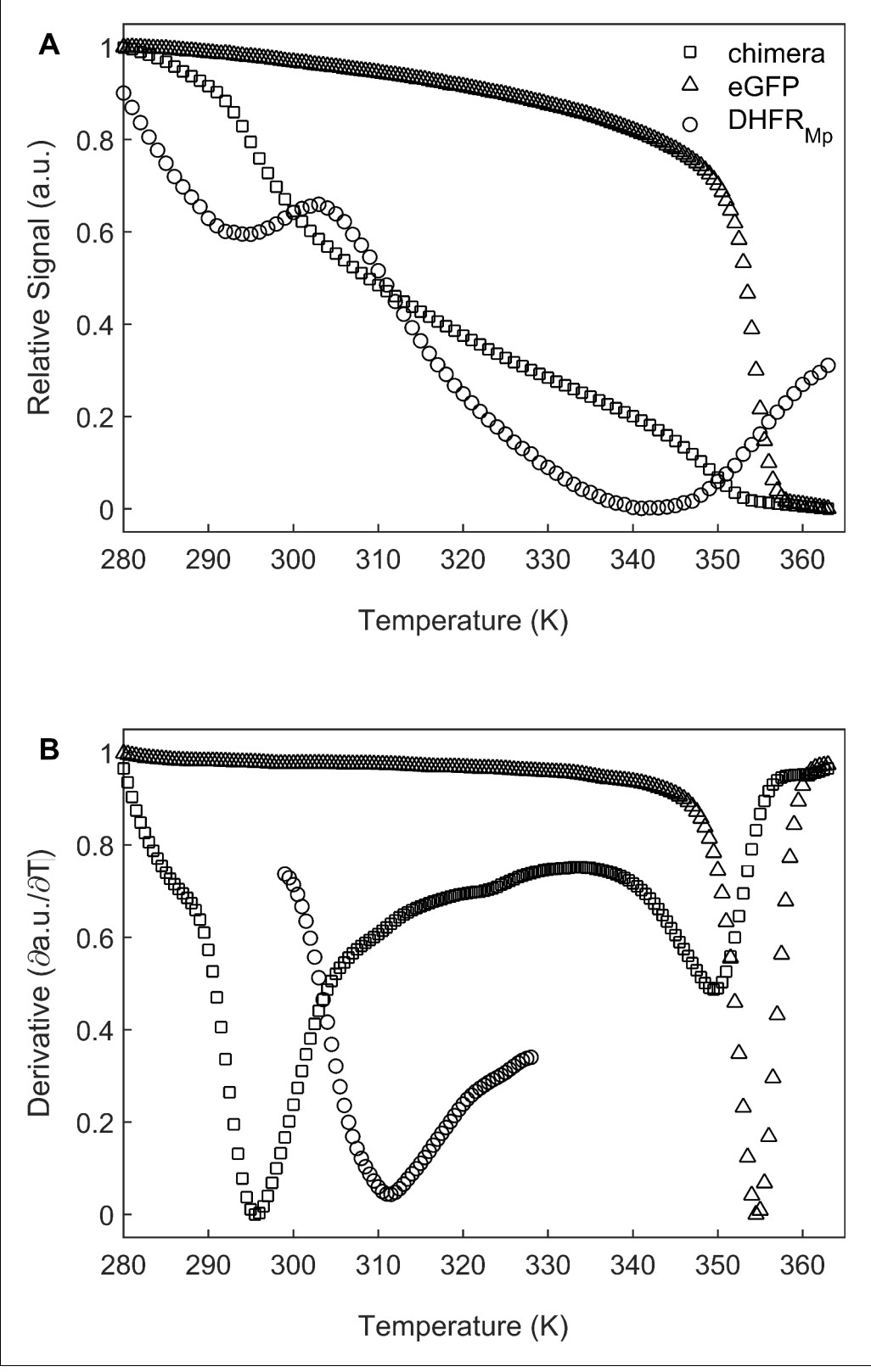

**Figure 2.** Melting temperature analysis of eGFP, DHFR$_{Mp}$ and their chimera. (**A**) DHFR$_{Mp}$, eGFP and a chimera of these proteins were subjected to temperature denaturation in bulk solution using a Real-Time PCR machine. Denaturation was monitored by measuring the change in fluorescence of the SYPRO Orange reagent. Each trace
*Figure 2 continued on next page*

*Figure 2 continued*

represents the average of at least six experiments. (**B**) Plots of the first derivatives of the denaturation curves as a function of temperature. Melting points were determined from these plots by using the StepOne Software v2.3. The online version of this article includes the following source data for figure 2:

**Source data 1.** Temperature melts of eGFP, DHFR$_{Mp}$and their chimera.

substrates bind only to the folded state, thereby shifting the equilibrium in its favor. In such a case, the rate constant associated with the lag phase should decrease with increasing substrate concentrations (*Vogt and Di Cera, 2012*). Alternatively, substrate-promoted folding of the DHFR$_{Mp}$ part of the chimera can also occur via a mechanism of induced fit in which case the rate constant associated with the lag phase should increase with increasing substrate concentrations. In other words, an increase in the rate constant of the lag phase with increasing substrate concentration can be due to substrate penetration or induced fit, whereas a decrease is evidence for conformational selection. Discriminating between these mechanisms can, therefore, be achieved by measuring the substrate concentration dependence of the rate constant of the lag phase.

The change in the absorbance at 340 nm due to the enzyme activity of the encapsulated or free chimera was, therefore, monitored in the presence of different concentrations of DHF and a fixed concentration of NADPH. The data were fitted to:

$$[P] = Vt + A\left(e^{-\lambda t} - 1\right) \tag{1}$$

where [P] is the product concentration (or the absorbance at 340 nm which is proportional to it), $V$ is the slope that corresponds to the linear steady-state velocity of the reaction and A and $\lambda$ are the respective amplitude and rate constant of the lag phase. *Equation 1* can be derived for the reaction scheme $E_U \rightleftharpoons E_F \rightleftharpoons ES$, where $E_U$, $E_F$ and ES designate the respective unfolded, folded and substrate-bound folded states of the protein (E), assuming that ligand binding is fast relative to the folding and unfolding steps. In such a case, the rate constant, $\lambda$, is given by *Vogt and Di Cera, 2012*:

$$\lambda = k_1 + \frac{k_{-1}k_{-2}}{k_2[S] + k_{-2}} = k_1 + \frac{k_{-1}}{K_a[S] + 1} \tag{2}$$

where $k_1$ and $k_{-1}$ are the respective folding and unfolding rate constants, $k_2$ and $k_{-2}$ are the respective substrate association and dissociation rate constants and $K_a = k_2/k_{-2}$ is the substrate association constant. Inspection of *Equation 2* shows that the value of the observed rate constant, $\lambda$, decreases with increasing substrate concentration as observed here in the case of the encapsulated chimera (*Figure 4A*). This finding indicates that the DHFR$_{Mp}$ part of the chimera is significantly destabilized in the cavity, but not in bulk solution (where the lag phase is absent), and that the substrates DHF and NADPH shift its equilibrium toward the folded state via a conformational selection mechanism. In agreement with this finding, temperature melts of the chimera in bulk solution show that both DHF and NADPH stabilize its DHFR$_{Mp}$ part (*Figure 4—figure supplement 1*). It should be noted that, in principle, a similar kinetic behavior would be observed for a scheme $E' \rightleftharpoons E_F \rightleftharpoons ES$, that is when the equilibrium is between the folded protein and a mis-folded inactive species, E' (instead of between the folded and unfolded species). This possibility is unlikely, however, because it was found that only 20% of human DHFR mis-folds inside the cavity of the non-cycling complex of single-ring GroEL with GroES (*Horst et al., 2007*). In such a case, the expected lag phase due to the mis-folded population would not be observed owing to the activity of the remaining DHFR, which does not mis-fold. Moreover, according to this model, the mis-folded state is much more stable than the folded state, which is in violation of Anfinsen's dogma that the native state is at the minimum free energy if the conditions in the cavity are assumed to favor folding.

Values of the folding and unfolding rate constants were obtained by fitting the plot of $\lambda$ as a function of substrate (DHF) concentration to *Equation 2* and found to be 1.4 ($\pm$0.4) x $10^{-4}$ and 7.3 ($\pm$0.8) x $10^{-3}$ sec, respectively (*Figure 4A,B*). Consequently, the stability of the cavity-confined fused DHFR$_{Mp}$, in the absence of the substrates DHF and NADPH, is 2.4 ($\pm$0.2) kcal mol$^{-1}$, that is the folded state is extremely destabilized so that it is less stable than the unfolded state. By contrast, the stability in bulk solution of fused DHFR$_{Mp}$, in the absence of the substrates DHF and NADPH, is −3.45 ($\pm$0.09) kcal mol$^{-1}$ (*Figure 4—figure supplement 2*). This value was determined by measuring

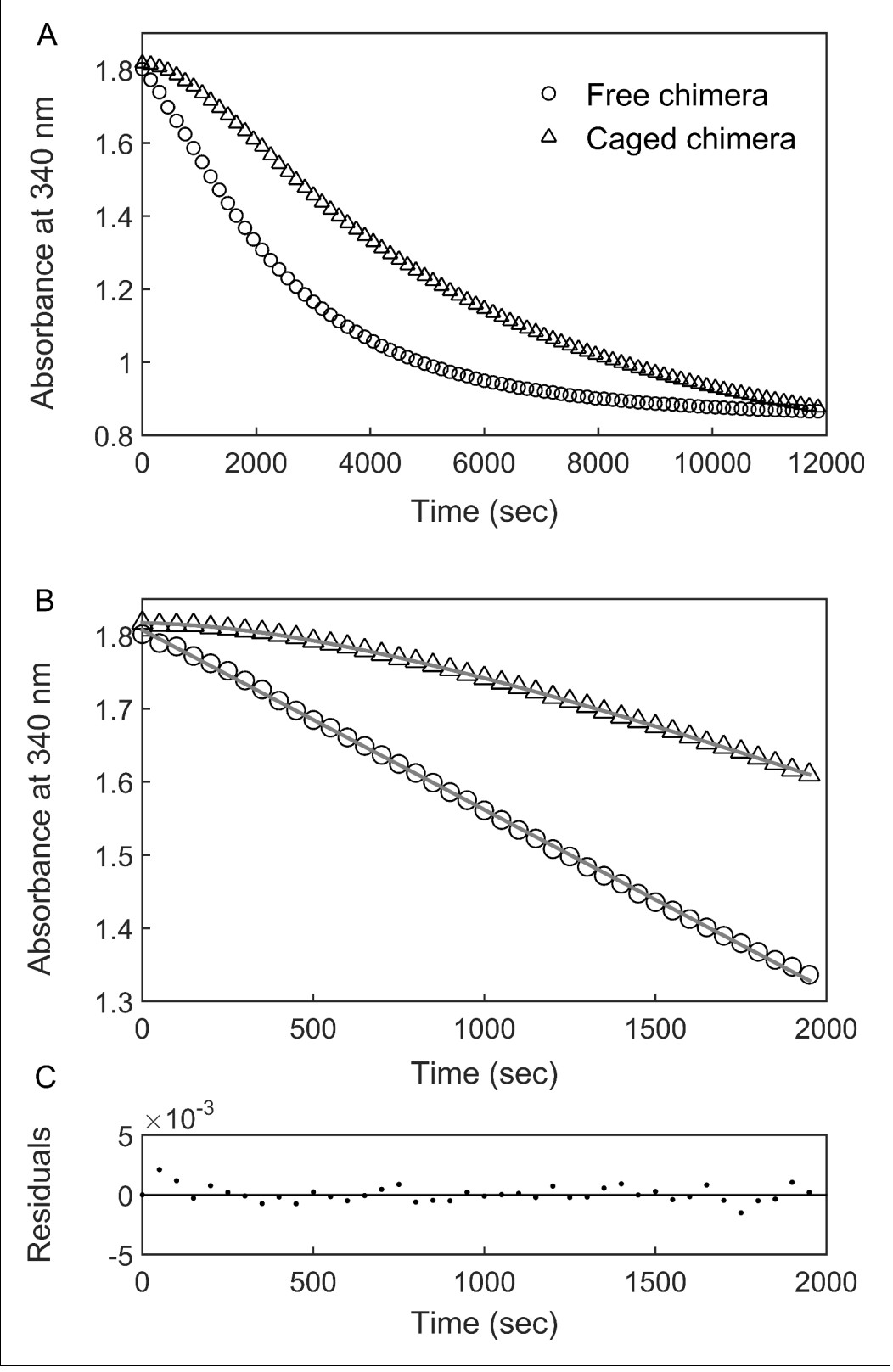

**Figure 3.** DHFR$_{Mp}$ activity of the GroE-encapsulated chimera shows an initial lag phase. (A) Representative traces are shown for the DHFR$_{Mp}$ activities of free (29 nM) and GroE-encapsulated (40 nM) chimera upon addition of 250 µM dihydrofolic acid and 300 µM NADPH. The reactions were monitored by measuring the decrease in absorbance at 340 nm as a function of time. An initial lag phase is observed in the case of the encapsulated

*Figure 3 continued on next page*

*Figure 3 continued*

chimera but not the free one. (B) The lag and linear phases in the data for the encapsulated chimera were fitted to *Equation 1*. (C) Good fits were obtained as indicated by plots of the residuals with random deviations about zero. See Materials and methods for additional details.

The online version of this article includes the following source data for figure 3:

**Source data 1.** Representative reaction progress curves of free and caged chimera in the presence of 250 µM DHF.

the stability of fused $DHFR_{Mp}$, in the presence of different concentrations of NADPH, in order to minimize aggregation that takes place in bulk solution, and then extrapolating to zero concentration of NADPH. Taken together, the results obtained here show that encapsulation in the GroEL cavity destabilizes the $DHFR_{Mp}$ part of the chimera by more than five kcal mol$^{-1}$ relative to bulk solution. Interestingly, analysis of the linear phase in the progress curves (e.g. in *Figure 3*) shows that the steady-state enzyme activity of $DHFR_{Mp}$ is also affected by encapsulation. The $K_m$ value of the encapsulated fused $DHFR_{Mp}$ for DHF is about 0.28 (±0.09) mM and, thus, about 10-fold higher than the value of 33 (±2) µM measured in bulk solution (*Figure 4C*), in agreement with a previous determination (*Xu et al., 2003*). The value of $k_{cat}$ may also be affected but these values are difficult to determine because of uncertainties regarding the active protein concentrations.

In order to test whether the destabilization of the cavity-confined fused $DHFR_{Mp}$ is due to its interaction with the cavity walls, we carried out fluorescence anisotropy decay measurements. Fits of the decay curves for the free and caged chimeras yielded similar rotational times of 22.85 (±0.30) ns and 25.40 (±0.45) ns, respectively (*Figure 5*). Likewise, the fits of the decay curves for free and caged eGFPs yielded similar rotational times of 14.38 (±0.20) ns and 15.90 (±0.25) ns, respectively (*Figure 5—figure supplement 1*), in agreement with previous work (*Striker et al., 1999*). The results indicate free mobility of the chimera and eGFP inside the GroEL football complex. It is, therefore, unlikely that the cavity-induced thermodynamic destabilization of the fused $DHFR_{Mp}$ is due to interactions with the cavity walls.

## Discussion

Our results show that protein stability in the chaperonin cage is reduced dramatically by more than 5 kcal mol$^{-1}$ compared to that in bulk solution. Given that steric confinement alone is expected to be stabilizing, our results indicate that protein destabilization in the cavity is likely to be due to hydrophobic and/or electrostatic effects. One possibility is that water in the cavity becomes more ordered in the presence of a substrate protein, thereby leading to a diminished hydrophobic effect. All-atom molecular dynamics simulations have shown that pairwise hydrophobic interactions are destabilized in hydrophobic nanopores when they contain water at bulk density (*Vaitheeswaran and Thirumalai, 2008*), as indicated by some evidence in the case of the GroEL cavity (*Franck et al., 2014*). The cavity walls of the 'football' complexes in our experiments are, however, charged and the extent of destabilization would, therefore, depend on many factors including the distribution of charges on the cavity walls, the cavity geometry and the presence of regions where the water structure is disrupted. Future simulations should test how these various factors affect pairwise interactions under confinement.

Similar to man-made machines, biological machines such as GroEL cycle between states that differ in their structural and functional properties. Substrate proteins first bind to GroEL in its apo state and can become destabilized through binding to its hydrophobic cavity walls (see, for example, *Libich et al., 2015*). ATP binding to the GroEL-protein substrate complex, which follows, can cause further unfolding due to a stretching force applied to the substrate protein upon ATP-binding-promoted conformational changes (*Weaver et al., 2017*). Next, protein substrates are encapsulated in the GroES-capped cavity for 1–10 s (*Bigman and Horovitz, 2019*) before being discharged into bulk solution. Here, we have succeeded in isolating the effect of this key step on substrate stability and found that it can also lead to protein destabilization. It is important to point out, however, that the magnitude (and possibly direction) of the effect is likely to depend on the protein substrate's size and other properties. Hence, future studies should employ the strategy developed here to examine other proteins.

Finally, it is important to note that our finding that the cavity environment is destabilizing supports the iterative annealing mechanism of action proposed for the chaperonin GroEL (*Todd et al., 1996*; *Thirumalai et al., 2020*). According to this mechanism, protein substrates undergo kinetic partitioning, during each reaction cycle of GroEL, between productive folding to the native state and mis-folding. The remaining fraction of mis-folded protein substrates are then rebound to GroEL and (partially) unfolded, thereby giving them new opportunity to fold correctly. Our results show that (partial) unfolding is achieved not only because of binding to the hydrophobic cavity of the apo state and ATP-promoted forced unfolding, as shown before by others, but also because it is strongly favored thermodynamically under the conditions in the GroES-capped cavity.

# Materials and methods

**Key resources table**

| Reagent type (species) or resource | Designation | Source or reference | Identifiers | Additional information |
|---|---|---|---|---|
| Recombinant DNA reagent | Pet21d plasmid for expressing DHFR$_{Mp}$ | Gift from Prof. S. Fleishman | | |
| Recombinant DNA reagent | Pet21a plasmid for expressing eGFP | PMID:29066625 | | |
| Strain, strain background (*Escherichia coli*) | *E. coli* Rosetta cells | Novagen | | |
| Strain, strain background (*Escherichia coli*) | *E. coli* BL21 cells | PMID:3537305 | | |
| Sequence-based reagent | TEV site insertion -forward | This work | PCR primer | 5'-ACTCGAGCACCACCAC CACCACCACTGA-3' |
| Sequence-based reagent | TEV site insertion -reverse | This work | PCR primer | 5'-GCTGTATAAGGGCA ACCTGTATTTTCAGGGCA CTCGAGCACCACC-3' |
| Sequence-based reagent | DHFR$_{Mp}$ cloning -forward | This work | RF cloning primer | 5'-CCTGTATTTTCAGGGCA TGATCGTAAGCATGAT TGCCGCACTGGCG-3' |
| Sequence-based reagent | DHFR$_{Mp}$ cloning -reverse | This work | RF cloning primer | 5'-GGTGGTGGTGGTGCTCG AGTTCACTCGAGTTTGACT CTTTCAAGTAGAC-3' |
| Chemical compound, drug | SYPRO Orange protein gel stain | Sigma | Cat#S5692 | Used at a dilution of 1:5000 |
| Software, algorithm | MATLAB 2015b | MathWorks | | |
| Other | HisTrap 5 ml columns | GE Healthcare | Cat#17-5248-02 | |
| Other | PD 10 desalting columns | GE Healthcare | Cat#17-0851-01 | |
| Other | Superdex 75 10/300 column | GE Healthcare | Cat#29-1487-21 | |
| Other | Superose 6 10/300 column | GE Healthcare | Cat#17-5172-01 | |
| Other | Q Sepharose column | GE Healthcare | Cat#17-1014-01 | |

## Construction of the chimera of DHFR$_{Mp}$ with eGFP

The Pet21a plasmid containing the gene for eGFP (with the mutation A206K that stabilizes its monomeric state) and an N-terminal His$_7$-tag (*Bandyopadhyay et al., 2017*) was amplified by PCR using the following respective forward and reverse primers:

5'-ACTCGAGCACCACCACCACCACCACTGA-3'

5'-GCTGTATAAGGGCAACCTGTATTTTCAGGGCACTCGAGCACCACC-3'. This amplification resulted in introducing a TEV protease cleavage site at the C-terminus of eGFP. The gene coding for DHFR$_{Mp}$ was then introduced at the 3' end of the TEV coding sequence by restriction-free cloning using the forward and reverse primers:

5'-CCTGTATTTTCAGGGCATGATCGTAAGCATGATTGCCGCACTGGCG-3'

5'-GGTGGTGGTGGTGCTCGAGTTCACTCGAGTTTGACTCTTTCAAGTAGAC-3', respectively. The final gene product codes for eGFP with an N-terminal His$_7$-tag fused at its C-terminus to DHFR$_{Mp}$ via a linker sequence containing a TEV protease cleavage site.

## Expression and purification of the chimera of DHFR$_{Mp}$ with eGFP

*E. coli* BL21 cells (*Studier and Moffatt, 1986*) harboring the Pet21d plasmid containing the gene for the chimera were inoculated into 250 ml 2 x TY with 100 µg/ml ampicillin and grown at 37°C until an O.D. of 0.6 was reached. Expression was induced by addition of 0.5 mM IPTG and growth was continued overnight at 16°C. The cells were then spun at 11,970 g for 20 min at 4°C, resuspended in 10% sucrose solution, spun again at 3,452 g for 30 min at 4°C and the pellet was stored at −80°C until purification. The pellet was resuspended in 30 ml of 50 mM Tris-HCl buffer (pH 7.5) containing 10 mM NaCl and 1 mM DTT (buffer A) with 8 M urea to prevent proteolysis. The cells were lysed using a French press (three passes) and sonication (4 cycles of 20 s sonication at 70% intensity and 60 s intervals). The lysate was then centrifuged at room temperature for 10 min at 24,610 g and again at 38,720 g for 30 min. The supernatant was diluted with buffer A to 6 M urea and loaded on a 140 ml Q-Sepharose column pre-equilibrated with buffer A containing 6 M urea. The column was then washed with 200 ml buffer A containing 6 M urea and with 150 ml of this buffer containing also 150 mM NaCl. A 5 ml HisTrap column, which was pre-equilibrated with buffer A containing 6 M urea, was connected downstream to the Q-Sepharose column and both columns were washed with 200 ml of buffer A with 1 M NaCl. The Q-Sepharose column was then removed and the HisTrap column was washed at room temperature with 75 ml of 50 mM Tris-HCl buffer (pH 7.5) containing 100 mM NaCl, 15 mM imidazole and 1 mM DTT (buffer B) with 5 M urea. Elution was carried out with a 75 ml linear gradient of 15 to 500 mM imidazole in buffer B with 5 M urea. Fractions were analyzed with SDS-PAGE and those containing the chimera were combined and concentrated to ~8 ml. The protein was then refolded at 4°C by mixing it with buffer B at a ratio of 1.5:98.5, respectively, during loading on a HisTrap 5 ml column. After loading, the column was washed with 25 ml buffer B and the protein was eluted with a 75 ml linear gradient from 15 to 500 mM imidazole in buffer B. Fractions containing the pure chimera were identified by SDS-PAGE and pooled and the buffer was then exchanged to 20 mM Hepes (pH 8.0) containing 100 mM KCl, 50 mM MgCl$_2$, 50 mM Na$_2$SO$_4$, 10 mM NaF, 1 mM BeSO$_4$ and 1 mM DTT (working buffer) using a PD 10 desalting column at 4°C. The concentration was determined from the absorption at 280 nm using an extinction coefficient of 50435 M$^{-1}$ cm$^{-1}$ and the protein was aliquoted and stored at −80°C for further use.

## Expression and purification of DHFR$_{Mp}$

Expression and purification of DHFR$_{Mp}$ were achieved as before (*Xu et al., 2003*) with the following changes. *E. coli* Rosetta cells harboring the Pet21d plasmid containing the DHFR$_{Mp}$ gene were inoculated into 7 L 2 x TY medium and grown at 37°C until an O.D. of 1 was reached. IPTG (0.5 mM) was then added to induce protein expression and the cells were grown at 37°C for another 5 hr. Harvesting was carried out by spinning the cells at 11,970 g for 20 min at room temperature and then resuspending them in 100 ml buffer B containing 10 µg/ml aprotinin, 5 µg/ml antipain, 5 µg/ml pepstatin, 5 µg/ml chymostatin, 10 µg/ml leupeptin, 50 µl EDTA-free protease inhibitor cocktail (Calbiochem), 1.5 µM PMSF, 6 µg/ml RNase A, 30 µg/ml DNase and 0.22 mg/ml folate. The cells were lysed by sonication (4 cycles of 20 s sonication at 70% intensity and 60 s intervals) and one passage through a French press at 1500 atm. The lysate was centrifuged at 4°C for 30 min at 24,610 g and then for 30 min at 38,720 g. The supernatant was applied to a 5 ml His-Trap column, which was then washed with 25 ml buffer B, 25 ml buffer B containing 1 mM ATP and 50 mM MgCl$_2$ and then with 25 ml buffer B again. DHFR$_{Mp}$ was eluted with a 50 ml linear gradient of 15 mM to 500 mM imidazole in buffer B. Fractions were analyzed with SDS-PAGE and those containing DHFR$_{Mp}$ were combined, concentrated and the buffer was exchanged to 50 mM Tris-HCl (pH 8.0) containing 250 mM NaCl, 5 mM EDTA, 5 mM β-mercaptoethanol (buffer C) using a PD-10 desalting column at 4°C. The protein

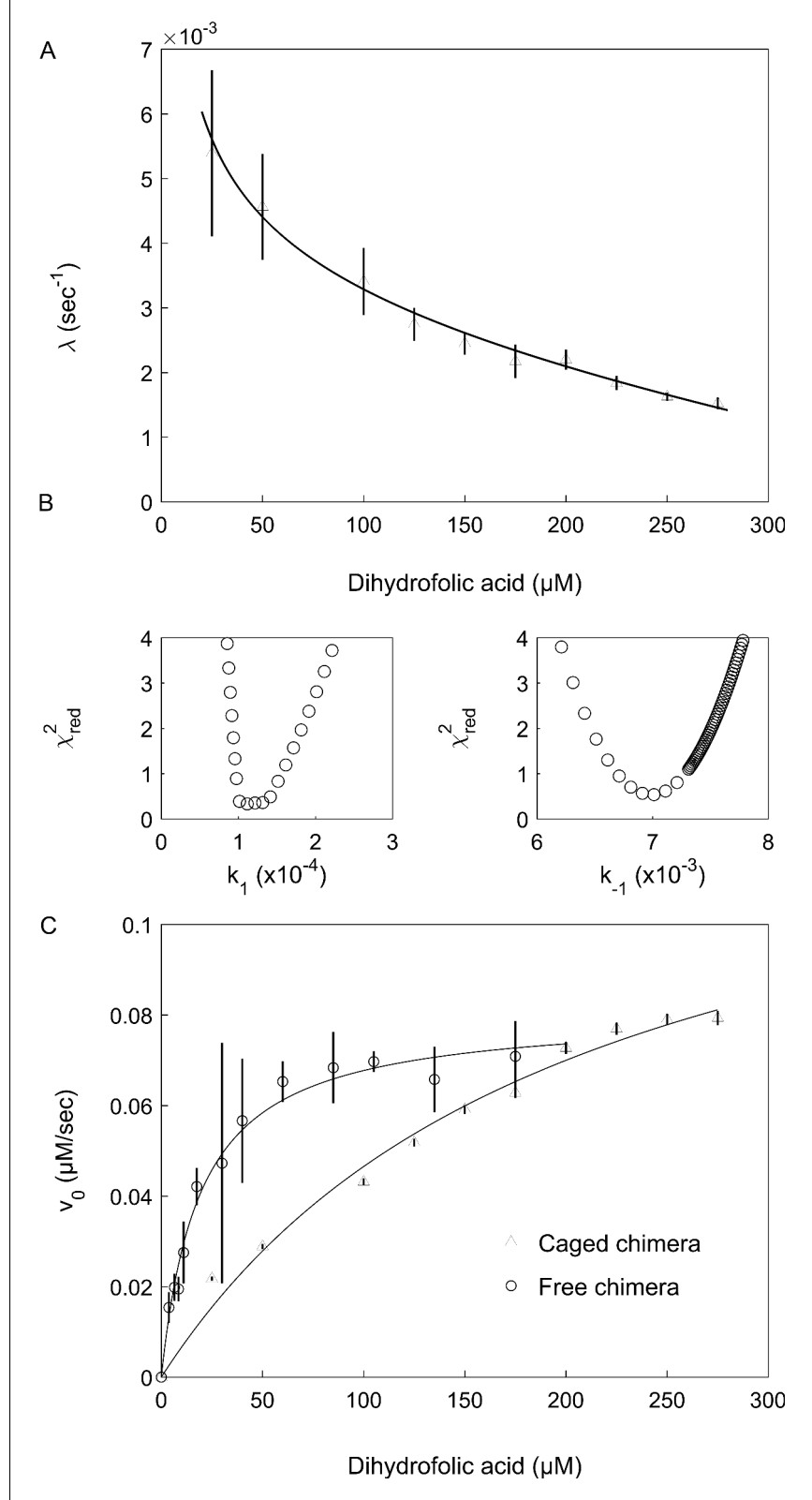

**Figure 4.** Encapsulation in the GroE cavity affects folding and enzyme activity of the DHFR_Mp part of the chimera. (A) Plot of the observed folding rate constant, $\lambda$, of the DHFR_Mp part of the chimera inside the cavity as a function of the DHF concentration. The data were fitted to *Equation 2*, thereby yielding estimates of the folding and unfolding rate constants. These experiments were carried out in duplicate. (B) Reduced $\chi^2$ surfaces for the folding

*Figure 4 continued on next page*

*Figure 4 continued*

and unfolding rate constants $k_1$ and $k_{-1}$, respectively. The relatively steep chi-squared surfaces suggest that the extracted parameters are well defined with relatively narrow confidence intervals. (C) Plot of the initial enzyme velocity of the DHFR$_{Mp}$ part of the chimera inside the cavity and in bulk solution, as a function of the DHF concentration. The data were fitted to the Michaelis-Menten equation. In the case of the encapsulated chimera, the initial enzyme velocities were obtained by fitting the data of decrease in absorption at 340 nm as a function of time to *Equation 1*. Error bars in both panels represent standard errors.

The online version of this article includes the following source data and figure supplement(s) for figure 4:

**Source data 1.** Reaction progress curves of caged chimera in the presence of different concentrations of DHF.
**Source data 2.** Reaction progress curves of free chimera in the presence of different concentrations of DHF.
**Figure supplement 1.** The melting temperature of DHFR$_{Mp}$ increases in the presence of its substrates.
**Figure supplement 1—source data 1.** Temperature melts of the chimera in the absence and presence of DHF or NADPH.
**Figure supplement 2.** Stability measurements of the DHFR$_{Mp}$ part of the chimera in bulk solution.
**Figure supplement 2—source data 1.** Free energies of folding of the DHFR part of the free chimera in the presence of different concentrations of NADPH.

---

was applied to a 5 ml methotrexate column, which was then washed with 20 ml buffer C. DHFR$_{Mp}$ was eluted with a linear gradient of 70 ml from 0 to 1 mM folate and 0.25 to 1 M NaCl in buffer C. Fractions were analyzed with SDS-PAGE and those containing DHFR$_{Mp}$ were combined, concentrated, mixed with 1 mM ATP and 50 mM MgCl$_2$ and then applied to a Superdex 75 10/300 gel-filtration column equilibrated with working buffer. Fractions were analyzed by SDS-PAGE and those containing pure DHFR$_{Mp}$ were combined, aliquoted and stored at −80°C. The protein concentration was determined using the Bradford assay and a BSA-based calibration curve.

## Measurements of melting temperatures of eGFP and DHFR$_{Mp}$

The melting temperatures of eGFP and DHFR$_{Mp}$ alone and in the chimera were measured by differential scanning fluorimetry using a StepOne real-time PCR instrument (Applied Biosystems). The protein (5 μM) alone or in the presence of 500 μM NADPH or dihydrofolate was mixed with 1X SYPRO Orange reagent (Sigma) in a 20 μl reaction volume of working buffer. The sample was heated from 4°C to 94°C with a temperature increment of 0.5°C every 45 s. The fluorescence of SYPRO Orange was measured after each temperature increment by exciting at 492 nm and measuring the emission at 610 nm. Analysis of the data was carried out using the StepOne software V2.3.

## Stability measurements of the DHFR$_{Mp}$ part of the chimera in bulk solution

The stability of the DHFR$_{Mp}$ part of the chimera was determined by measuring the fluorescence emission at 330 nm upon excitation at 280 nm (using an ISS PC1 fluorimeter with excitation and emission bandwidths of 16 nm) as a function of GuHCl concentration at different fixed concentrations of NADPH. Measurements were made at 23°C for samples containing about 1.4 μM of the chimera in working buffer after incubation for 10 min at the same temperature. The concentration of the stock solution of GuHCl was determined by measuring the refraction index at 23°C. The data were fitted using the following equation:

$$F = \frac{F_U^0 + a[D] + \left(F_N^0 + b[D]\right)e^{\frac{-\Delta G^0 + m[D]}{RT}}}{1 + e^{\frac{-\Delta G^0 + m[D]}{RT}}}$$

(3)

where $\Delta G^0$ is the free energy of unfolding in the absence of denaturant, [D] is the GuHCl concentration, m is the GuHCl-concentration dependence of the free energy of unfolding ($m = \frac{\partial \Delta G}{\partial [D]}$), T is the temperature and R is the gas constant. The fluorescence of the native (N) and denatured (U) states are expressed as linear functions of the GuHCl concentration with slopes of a and b, respectively. This analysis is based on the assumption that the melting curves reflect the denaturation of only the DHFR$_{Mp}$ part of the chimera, which is justified since eGFP is very stable under our conditions and the fluorescence of its single tryptophan residue changes very little (and in a linear fashion) as a function of GuHCl concentration. The values of $\Delta G^0$ obtained from the fits to *Equation 3* were then

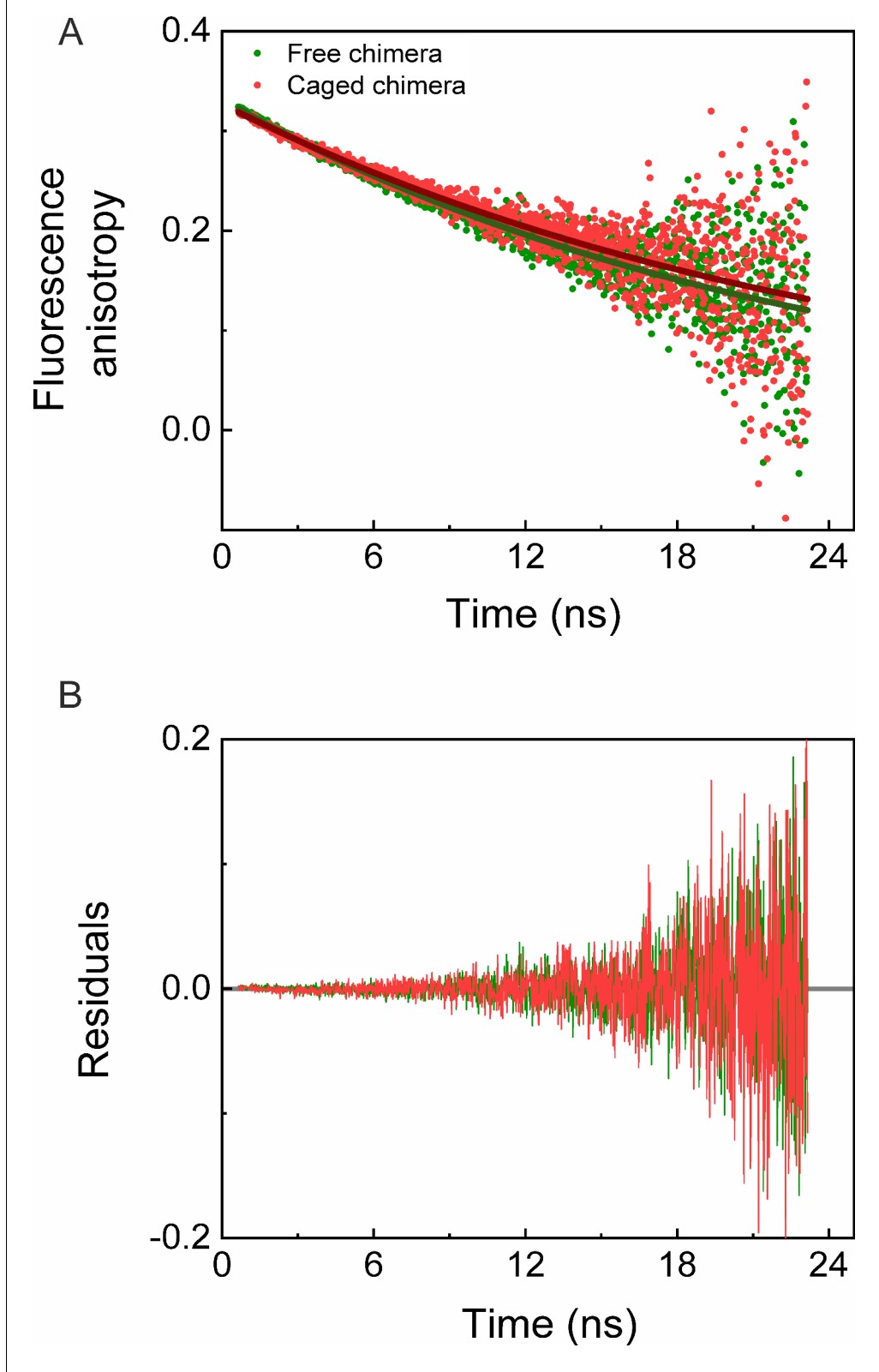

**Figure 5.** Time-resolved fluorescence anisotropy measurements of free and GroEL-caged chimera. (**A**) Fluorescence anisotropy decay curves of free (green) and caged (red) chimeras were measured as described in the Materials and methods. Fits of the decay curves for free and caged chimera to a single exponential (green and red solid lines, respectively) yielded rotational times of 22.85 (±0.30) ns and 25.40 (±0.45) ns, respectively. The results

*Figure 5 continued on next page*

*Figure 5 continued*

indicate no restriction on the mobility of the chimera inside the GroEL football complex. Each experiment was carried out in duplicate. (**B**) Plots of the residuals for the fits in panel A.

The online version of this article includes the following source data and figure supplement(s) for figure 5:

**Source data 1.** Time-resolved fluorescence anisotropy measurements of free and caged eGFP.
**Figure supplement 1.** Time-resolved fluorescence anisotropy measurements of free and caged eGFP.
**Figure supplement 1—source data 1.** Time-resolved fluorescence anisotropy measurements of free and caged chimera.

plotted as a function of the concentration of NADPH and the data were fitted using the following equation:

$$\Delta G = \Delta G^0 - RTln(1 + [S]/K_m) \tag{4}$$

where [S] is the concentration of NADPH and $K_m$ is its Michaelis-Menten constant of 13 μM (*Evans et al., 2010*). Here, $\Delta G^0$ is the free energy of unfolding in the absence of both GuHCl and NADPH.

## Encapsulation of protein substrates in GroE footballs

Encapsulation of eGFP and the chimera in the GroEL-GroES$_2$ football was performed using different methods of denaturation. In the case of eGFP, the substrate (7.5 nmol) was denatured in 200 μl of 60 mM HCl in a siliconized test-tube. The denatured eGFP was then added slowly to 20 ml of 50 mM Tris-HCl buffer (pH 7.5) containing 100 mM KCl, 50 mM MgCl$_2$, 1 mM DTT, 0.05 μM GroEL oligomer and 0.125 μM GroES oligomer (folding buffer) and left for 40 min at room temperature with mild stirring. Football assembly was initiated by adding 2 ml containing 550 mM Na$_2$SO$_4$, 110 mM NaF, 11 mM BeSO$_4$ and 11 mM ATP (activation mix) to the folding buffer. The mixture was then stirred for 15 min at room temperature, concentrated and fractionated using a Sepharose 6 (10/300) gel-filtration column in working buffer. Fractions were analyzed by SDS-PAGE and those containing GroE-encapsulated eGFP were kept for further work. In the case of the chimera, a binary complex was first formed by incubating 9 nmol of the native chimera with ~5 nmol of GroEL oligomer in 300 μl of working buffer overnight at room temperature. The mixture was then added to a 200 μl solution containing 135 mM Na$_2$SO$_4$, 25 mM NaF, 2.5 mM BeSO$_4$, 2.5 mM ATP and ~10 nmol of GroES oligomer, stirred for 15 min at room temperature and applied to a Superose 6 10/300 gel-filtration column.

## DHFR activity assays

Activity assays were performed at 23°C in working buffer using an Infinite M200pro (TECA Group Ltd.) plate reader. Reactions were initiated by mixing 190 μl of the protein with 10 μl of NADPH (350 μM final concentration) and different concentrations of DHF. The reaction progress was followed by measuring the decrease in absorption at 340 nm as a function of time. The chimera concentrations were verified from measurements of the fluorescence emission at 509 nm upon excitation at 488 nm (with bandwidths of 4, 8 or 16 nm depending on the protein concentration) and using an eGFP-based calibration curve. Analysis and fitting of the data were performed using MatLab 2015b software.

## Time-resolved fluorescence anisotropy measurements

Fluorescence anisotropy decay curves of the target proteins were measured using a MicroTime200 fluorescence microscope (PicoQuant). Samples of eGFP and the chimera were diluted to 50–100 nM in the working buffer containing 0.01% TWEEN 20 and then loaded into a flow cell pre-coated with a lipid bilayer (*Mazal et al., 2019*). Molecules were excited with a 485 nm diode laser pulsed at a repetition rate of 20 MHz and with a power of 10 μW. Emitted photons were divided based on their polarization using a polarizing beam splitter cube, followed by filtration using band-pass filters (520/35 nm, BrightLine). Photon arrival times relative to the excitation pulse were registered with a resolution of 16 ps using two single-photon avalanche photo-diodes detectors (Excelitas SPCM-AQR-14-TR) coupled to a time-correlated single-photon counting module (HydraHarp 400, PicoQuant). The parallel and perpendicular fluorescence decays were constructed from the data and background

corrected. Fluorescence anisotropy decays were then calculated using the following relation: $r(t) = \frac{I_{\parallel(t)} - GI_{\perp(t)}}{I_{\parallel(t)} + 2GI_{\perp(t)}}$, where $I_{\parallel(t)}$ and $I_{\perp(t)}$ are the time-dependent fluorescence intensities of the parallel and perpendicular components, and G is the polarization sensitivity factor (whose value was determined to be 1). Fluorescence anisotropy decays were fitted to a single exponential function to obtain the rotational correlation times of the proteins.

## Acknowledgements

This work was supported by grant 2015170 of the US-Israel Binational Science Foundation, the Minerva Foundation with funding from the Federal German Ministry for Education and Research and the Kimmelman Center for Biomolecular Structure and Assembly. AH is an incumbent of the Carl and Dorothy Bennett Professorial Chair in Biochemistry.

## Additional information

### Funding

| Funder | Grant reference number | Author |
|--------|------------------------|--------|
| United States-Israel Binational Science Foundation | 2015170 | Amnon Horovitz |
| Minerva Foundation | | Amnon Horovitz |

The funders had no role in study design, data collection and interpretation, or the decision to submit the work for publication.

### Author contributions

Ilia Korobko, Data curation, Formal analysis, Investigation; Hisham Mazal, Data curation, Formal analysis; Gilad Haran, Formal analysis, Writing - review and editing; Amnon Horovitz, Conceptualization, Formal analysis, Funding acquisition, Writing - original draft, Project administration

### Author ORCIDs

Hisham Mazal http://orcid.org/0000-0002-2071-9552
Gilad Haran http://orcid.org/0000-0003-1837-9779
Amnon Horovitz https://orcid.org/0000-0001-7952-6790

### Decision letter and Author response

Decision letter https://doi.org/10.7554/eLife.56511.sa1
Author response https://doi.org/10.7554/eLife.56511.sa2

## Additional files

### Supplementary files

• Transparent reporting form

### Data availability

All data generated or analyses during this study are included in the manuscript file.

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

## Appendix 1

We consider the following three-state reversible reaction:

$$A \underset{k_{-1}}{\overset{k_1}{\rightleftharpoons}} B \underset{k_{-2}}{\overset{k_2}{\rightleftharpoons}} C \tag{A1}$$

The rates of changes in the concentrations of A, B and C are given by:

$$d[A]/dt = -k_1[A] + k_{-1}[B] \tag{A2}$$

$$d[B]/dt = k_1[A] - (k_{-1} + k_2)[B] + k_{-2}[C] \tag{A3}$$

$$d[C]/dt = k_2[B] - k_{-2}[C] \tag{A4}$$

Setting d[A]/dt = -λ[A], d[B]/dt = -λ[B] and d[C]/dt = -λ[C] (where λ >0), we can express the above three rate equations in the following matrix form:

$$\begin{bmatrix} k_1 - \lambda & -k_{-1} & 0 \\ -k_1 & k_{-1} + k_2 - \lambda & -k_{-2} \\ 0 & -K_2 & k_{-2} - \lambda \end{bmatrix} \begin{bmatrix} [A] \\ [B] \\ [C] \end{bmatrix} = 0 \tag{A5}$$

Given that the determinant of the above matrix must be equal to zero, one obtains:

$$\lambda^2 - \lambda(k_1 + k_{-1} + k_2 + k_{-2}) + (k_1 k_2 + k_1 k_{-2} + k_{-1} k_{-2}) = 0 \tag{A6}$$

*Equation A6* has two solutions, $\lambda_1$ and $\lambda_2$, given by:

$$\lambda_1 + \lambda_2 = k_1 + k_{-1} + k_2 + k_{-2} \tag{A7}$$

$$\lambda_1 \lambda_2 = k_1 k_2 + k_1 k_{-2} + k_{-1} k_{-2} \tag{A8}$$

We can now consider two special cases described by Scheme A1, induced fit and conformational selection, both of which involve ligand binding and a conformational change. In the case of induced fit, ligand binding precedes and induces the conformational change. In the case of conformational selection, the ligand binds to one of several pre-existing conformational states and the conformational change, therefore, precedes ligand binding. In accordance with Scheme A1, the induced fit and conformational selection models can be described, respectively, as follows:

$$E + S \underset{k_{-1}}{\overset{k_1}{\rightleftharpoons}} ES \underset{k_{-2}}{\overset{k_2}{\rightleftharpoons}} E'S \tag{A9}$$

$$E \underset{k_{-1}}{\overset{k_1}{\rightleftharpoons}} E' + S \underset{k_{-2}}{\overset{k_2}{\rightleftharpoons}} E'S \tag{A10}$$

where E stands for the protein (enzyme), S for the substrate (ligand) and prime designates the conformational change. Assuming that ligand binding is much faster than the conformational change, one can rewrite *Equation A7* for induced fit, as follows:

$$\lambda_1 = k_1[S] + k_{-1} \tag{A11}$$

It follows by combining *Equations A8 and A11* that the observed rate constant is given by:

$$\lambda_2 = \frac{k_1 k_2 [S]}{k_1 [S] + k_{-1}} + k_{-2} \tag{A12}$$

In the case of conformational selection, one can rewrite *Equation A7*, as follows:

$$\lambda_2 = k_2 [S] + k_{-2} \tag{A13}$$

The observed rate constant in the case of conformational selection is, therefore, given by:

$$\lambda_2 = \frac{k_{-1} k_{-2}}{k_2 [S] + k_{-2}} + k_1 \tag{A14}$$

*Equation A14* corresponds to *Equation 2* in the main text. Inspection of *Equations A12 and A14* shows that, in the case of induced fit, the value of the observed rate constant increases with increasing substrate concentration whereas, in the case of conformational selection, it decreases. The decrease, in the case of conformational selection, is due to the fact that the reaction is essentially irreversible at high [S] and $\lambda_2$, therefore, tends towards $k_1$ whereas at low [S] the reverse rate becomes more significant and $\lambda_2$ tends towards $k_1 + k_{-1}$. In other words, the contribution of the reverse rate, $k_{-1}$, is weighted by the relative reverse and forward fluxes of the second step ($k_{-2}/(k_2[S] +k_{-2})$).

In this work, E and E′ correspond to the unfolded and folded states of the protein. In the case of conformational selection, the formation of E′S (i.e. the complex of the folded protein with substrate) is governed by the rate constant $\lambda_2$ (*Equation A14*). The concentration of E′S is therefore given by:

$$[E'S] = [E'S]_{ss} \left( 1 - exp^{-\lambda_2 t} \right) \tag{A15}$$

where [E′S]$_{ss}$ is the steady-state concentration of E′S (i.e. under conditions where the substrate concentration does not become limiting) and assuming that $k_{cat}$ can be neglected here. Product (P) formation is, therefore, given by:

$$\frac{d[P]}{dt} = k_{cat} [E'S] = k_{cat} [E'S]_{ss} \left( 1 - exp^{-\lambda_2 t} \right) \tag{A16}$$

Integration of *Equation A16* yields:

$$[P] = k_{cat} \left[ E'S \right]_{ss} t + \frac{k_{cat} [E'S]_{ss}}{\lambda_2} \left( exp^{-\lambda_2 t} - 1 \right) = Vt + A \left( exp^{-\lambda_2 t} - 1 \right) \tag{A17}$$

where $k_{cat}$[E′S]$_{ss}$ corresponds to the steady-state enzyme velocity, V, and A is a constant equal to $\frac{k_{cat} [E'S]_{ss}}{\lambda_2}$. *Equation A17* corresponds to *Equation 1* in the main text, which was used to fit the data of product formation vs. time as in *Figure 3*.

