## [Decision Letter]

Thank you for submitting your article "Measuring protein stability in the GroEL chaperonin cage reveals massive destabilization" for consideration by *eLife*. Your article has been reviewed by three peer reviewers, one of whom is a member of our Board of Reviewing Editors, and the evaluation has been overseen by David Ron as the Senior Editor.

The reviewers have discussed the reviews with one another and the Reviewing Editor has drafted this decision to help you prepare a revised submission.

Summary:

There is enthusiasm for the innovative approach to learn how the interior of the GroEL chaperonin chamber influences protein stability. The reviewers felt that this study potentially addresses a number of important questions about substrate confinement and GroEL mechanism. However, the implications of the work were not clearly articulated and consequently the impact of your study to the readership may not be as significant as it could be. Moreover, all reviewers felt that significantly more detail is required for a careful evaluation of the manuscript. Although no new experiments are required, a thoughtful major revision would be necessary where additional details are included. In a suitably revised version it is requested that the following points be considered:

Essential revisions:

1) Can the authors provide more evidence that the substrate is unfolded inside the cavity prior to addition of DHF/NADPH. I presume that the temperature used is close to the melting temperature of the substrate (but I could not find it listed – please put this in the text), but the melting temp of the substrate is measured outside of the cage so it is difficult to make the argument about what is happening inside the cavity based simply on temperatures.

2) A related question to 1. How can the authors be certain that the transition E to E' is actually U to N? Could this not be some binding incompetent state to one that is competent, but not necessarily U to N? And if this is the case then are the authors measuring folding/unfolding from k_1_ and k_-1_?

3) Could the authors please provide chi-sq surfaces of k_1_ and k_-1_, as the data from which they are extracted are quite noisy.

4) While I could derive the equations for induced fit and conformational exchange (although the authors may wish to see if there is a sign error in the eigenvalues – negative of what they are listed), I could not understand Equation 1. Please derive this equation as well, as it is critical to understanding the paper, making the link between the derivation that the authors have given for the lamda in appendix and how this explicitly relates to product formation as per Equation 1.

5) A major concern is that the authors use a species of GroEL/ES that is favored by beryllium fluoride with two GroES rings bound to the GroEL tetradecamer. This state is meant to mimic a state after a substrate is released into the cavity and in general would be assumed to be favored to fold. However, the relevance of the state the authors have used, apparently because it is stable and does not allow leakage of the trapped protein, to the mechanism of GroEL/ES is not argued convincingly. So the goal of the study is not clear: is it to show that entrapment in a small cavity can have destabilizing effects on a protein? or is it to contribute to understanding of the GroEL mechanism?

6) It was shown that eGFP doesn't leak from the GroEL/ES bullet, but not the chimera. Why was this not tested?

7) The paper is very brief and lacks a clear conclusion about the significance of the observations. The authors refer to the iterative annealing mechanism without any description or explanation of how their result relates.

8) It was unclear what the authors did differently that enabled them to confine the chimera in the cavity where previous attempts failed. Perhaps, the authors can spell this out.

9) The attachment of eGFP does destabilize DHFR a lot and eGFP as well. One assumes that the results in Figure 2 and subsection “Design and construction of a model protein substrate” are in bulk, although this was not spelt out explicitly. Please confirm.

10) The argument, in the last paragraph of the subsection “Folding of the DHFR_Mp_ part of the chimera in the GroEL cavity”, that the decay of fluorescence anisotropy of the two proteins are similar in the cavity and bulk, which is the basis for ruling out interaction with the wall being the cause of the destabilization. This is reasonable. Two naïve questions:

i) The decay simply suggests that the overall shape is the same but cannot rule out weak interactions. The decrease in the amplitude is not very large in either situations. Does this mean a "preferred" orientation?

ii) On time scales greater that about 18 ns (Figure 5) there is more dispersion (conformational variability?) in the caged construct. Is there an explanation or does this not matter?

11) The authors conclude that the effect could be due to diminished hydrophobic effect in the confined space. The authors might want to know that this was established in MD simulations (PNAS 105: 17636 (2008)) by calculating the potential of mean force between a few amino acid side chains (polar, charged, and hydrophobic). Among these it was clear that interactions between hydrophobic residues were diminished in a cylindrical cavity, supporting the conclusion here, whereas others are more nuanced. The authors realize that the results that the magnitude of the effects could be substrate specific. But given that the hydrophobic interactions might be the determining factor (assuming it holds) do they the authors think that the results are very general. In other words, destabilization should be the norm, which is in accord with the expectation of the iterative annealing mechanism.

---

## [Author Response]

Essential revisions:1) Can the authors provide more evidence that the substrate is unfolded inside the cavity prior to addition of DHF/NADPH. I presume that the temperature used is close to the melting temperature of the substrate (but I could not find it listed – please put this in the text), but the melting temp of the substrate is measured outside of the cage so it is difficult to make the argument about what is happening inside the cavity based simply on temperatures.

The temperature used, which is indeed close to the melting temperature in bulk solution, was stated in the Materials and methods of the original version and in the revised paper is mentioned also in the main text. Please see response below for more.

2) A related question to 1. How can the authors be certain that the transition E to E' is actually U to N? Could this not be some binding incompetent state to one that is competent, but not necessarily U to N? And if this is the case then are the authors measuring folding/unfolding from k_1_ and k_-1_?

This is an excellent point. Formally, a similar kinetic behavior would be observed for a scheme E′⇄EF⇄ES, i.e. when the equilibrium is between the folded protein and a mis-folded inactive species, E’. This possibility is unlikely, however, because it was found that only 20% of human DHFR mis-folds inside the cavity of the non-cycling complex of single-ring GroEL with GroES (Horst et al., 2007). In such a case, the expected lag phase due to the mis-folded population would not be observed owing to the activity of the remaining DHFR, which does not mis-fold. Moreover, according to this model, the mis-folded state is much more stable than the folded state, which is in violation of Anfinsen’s dogma that the native state is at the minimum free energy if the conditions in the cavity are assumed to favor folding. This discussion was added to the paper.

3) Could the authors please provide chi-sq surfaces of k_1_ and k_-1_, as the data from which they are extracted are quite noisy.

We thank the reviewer for this suggestion. The revised version contains a reduced chi-square surface analysis, which validates our estimates of k_1_ and k_-1_. We also reanalyzed the data, which were used to extract the values of lamda, in order to ensure that the linear phase is fully included until substrate depletion sets in and that deviations from linearity owing to substrate depletion are excluded. The reanalysis led to small changes in some of the lamda values and the impact on the values of the k_1_ and k_-1_ and the free energy of folding in the cavity is also small.

4) While I could derive the equations for induced fit and conformational exchange (although the authors may wish to see if there is a sign error in the eigenvalues – negative of what they are listed), I could not understand Equation 1. Please derive this equation as well, as it is critical to understanding the paper, making the link between the derivation that the authors have given for the lamda in appendix and how this explicitly relates to product formation as per Equation 1.

The appendix now also contains the derivation of Equation 1 and the link to λ. The apparent sign error was also fixed. The confusion was due to defining -λ as λ.

5) A major concern is that the authors use a species of GroEL/ES that is favored by beryllium fluoride with two GroES rings bound to the GroEL tetradecamer. This state is meant to mimic a state after a substrate is released into the cavity and in general would be assumed to be favored to fold. However, the relevance of the state the authors have used, apparently because it is stable and does not allow leakage of the trapped protein, to the mechanism of GroEL/ES is not argued convincingly. So the goal of the study is not clear: is it to show that entrapment in a small cavity can have destabilizing effects on a protein? or is it to contribute to understanding of the GroEL mechanism?

The reviewer is correct in stating that it has often been assumed that folding is favored in the cavity of GroEL bound to ATP and GroES but this has never been measured. By contrast, the effects of other states such as apo GroEL and ATP-bound GroEL are better understood. Our goal was to isolate the effect of this state from the effects of other states, which co-exist when the system is cycling. Surprisingly, we found that encapsulation in the cavity of GroEL bound to ATP and GroES can lead to massive destabilization. A new paragraph was added to the Conclusions to clarify this point and relate it to GroEL’s mechanism.

6) It was shown that eGFP doesn't leak from the GroEL/ES bullet, but not the chimera. Why was this not tested?

Given that eGFP doesn’t leak it seemed unnecessary to test whether a larger protein containing eGFP leaks. Moreover, we had tested before whether two other proteins, the p53 core domain (22.4 kDa) and the chimera of EnHD with eGFP (42.9 kDa), leak and found that neither does. We’ve added this information to the paper.

7) The paper is very brief and lacks a clear conclusion about the significance of the observations. The authors refer to the iterative annealing mechanism without any description or explanation of how their result relates.

A brief but hopefully clear description of the iterative annealing mechanism and how our result relates to it has been added to the Conclusions.

8) It was unclear what the authors did differently that enabled them to confine the chimera in the cavity where previous attempts failed. Perhaps, the authors can spell this out.

In the subsection “Identifying conditions for minimizing escape of encapsulated substrates”, we now state explicitly that we did not encapsulate the substrate in the cavity formed by single-ring GroEL with GroES, which was believed to be non-leaky.

9) The attachment of eGFP does destabilize DHFR a lot and eGFP as well. One assumes that the results in Figure 2 and subsection “Design and construction of a model protein substrate” are in bulk, although this was not spelt out explicitly. Please confirm.

These results are indeed for bulk solution as stated explicitly in the text and legend to Figure 2 of the revised version,

10) The argument, in the last paragraph of the subsection “Folding of the DHFR_Mp_ part of the chimera in the GroEL cavity”, that the decay of fluorescence anisotropy of the two proteins are similar in the cavity and bulk, which is the basis for ruling out interaction with the wall being the cause of the destabilization. This is reasonable. Two naïve questions:i) The decay simply suggests that the overall shape is the same but cannot rule out weak interactions. The decrease in the amplitude is not very large in either situations. Does this mean a "preferred" orientation?

We think that weak interactions would decrease rotational diffusion and therefore lead to longer anisotropy decay time. Therefore the similarity of the decays suggests that such interactions are not significant.

ii) On time scales greater that about 18 ns (Figure 5) there is more dispersion (conformational variability?) in the caged construct. Is there an explanation or does this not matter?

The ‘dispersion’ is not conformational variability but noise. This noise occurs because the fluorophore has to a large extent decayed already, so the polarization signal is a difference between two very small signals divided by a sum of two small signals, which means a lot of noise. This has nothing to do with the conformational dynamics, though, and in fact the noise in the green and red signals looks similar.

11) The authors conclude that the effect could be due to diminished hydrophobic effect in the confined space. The authors might want to know that this was established in MD simulations (PNAS 105: 17636 (2008)) by calculating the potential of mean force between a few amino acid side chains (polar, charged, and hydrophobic). Among these it was clear that interactions between hydrophobic residues were diminished in a cylindrical cavity, supporting the conclusion here, whereas others are more nuanced. The authors realize that the results that the magnitude of the effects could be substrate specific. But given that the hydrophobic interactions might be the determining factor (assuming it holds) do they the authors think that the results are very general. In other words, destabilization should be the norm, which is in accord with the expectation of the iterative annealing mechanism.

A discussion of this issue and its relevance to our work was added to the Conclusions. We thank the reviewer for pointing out this reference, which we cite in the revised paper.